# Panchromatic and Multispectral Image Fusion via Alternating Reverse Filtering Network

**Keyu Yan**[1,2]*, **Man Zhou**[1,2]*, **Jie Huang**[2], **Feng Zhao**[2], **Chengjun Xie**[1], **Chongyi Li**[3],
**Danfeng Hong**[4†]

[1]Hefei Institute of Physical Science, Chinese Academy of Sciences, China
[2]University of Science and Technology of China, China
[3]Nanyang Technological University, Singapore
[4]Aerospace Information Research Institute, Chinese Academy of Sciences, China

## Abstract

Panchromatic (PAN) and multi-spectral (MS) image fusion, named Pan-sharpening, refers to super-resolve the low-resolution (LR) multi-spectral (MS) images in the spatial domain to generate the expected high-resolution (HR) MS images, conditioning on the corresponding high-resolution PAN images. In this paper, we present a simple yet effective *alternating reverse filtering network* for pan-sharpening. Inspired by the classical reverse filtering that reverses images to the status before filtering, we formulate pan-sharpening as an alternately iterative reverse filtering process, which fuses LR MS and HR MS in an interpretable manner. Different from existing model-driven methods that require well-designed priors and degradation assumptions, the reverse filtering process avoids the dependency on pre-defined exact priors. To guarantee the stability and convergence of the iterative process via contraction mapping on a metric space, we develop the learnable multi-scale Gaussian kernel module, instead of using specific filters. We demonstrate the theoretical feasibility of such formulations. Extensive experiments on diverse scenes to thoroughly verify the performance of our method, significantly outperforming the state of the arts.

## 1   Introduction

Multispectral images are widely used in various fields such as resource monitoring [1], environmental protection [2, 3] and ecological monitoring [4]. However, due to the hardware limitations of multispectral sensors, multispectral images usually lack high spatial resolution [5, 6]. In contrast, high-resolution panchromatic (PAN) images that contain rich spatial details of the same scene are easy to obtain. Therefore, pan-sharpening, generating high-resolution multispectral images by fusing panchromatic images with low-resolution multispectral images, has become an important issue in the field of remote sensing [7, 8].

Many efforts have been made to solve the pan-sharpening problem, which can be generally divided into two large groups: traditional pan-sharpening methods and deep learning-based methods [9, 10]. Traditional pan-sharpening methods usually require strict assumptions of multispectral image degradation via prior knowledge. Otherwise, the inexact assumptions may cause system model error. For example, both the assumptions established by component substitution methods [11, 12] and multi-resolution analysis methods [13, 14] focus on the relationship between PAN and HR MS, which is destined to make these methods prone to spectral distortion. Unlike the above-mentioned traditional approaches, variational optimization approaches consider the relationship among HR MS, LR MS

---

*Co-first authors contributed equally, † corresponding author.

36th Conference on Neural Information Processing Systems (NeurIPS 2022).

and PAN, and then construct the energy function based on well-designed priors [15, 16]. However, the methods of this kind inflict high computational burden, restricting their practical applications.

Most recently, deep learning exhibited outstanding performance in the field of remote sensing images [17, 18]. Undoubtedly, the deep learning-based (DL) methods become a newly developed category to solve pan-sharpening [19–22]. Unfortunately, the pan-sharpening networks commonly lack the interpretability in theirs designs, which limits their performance. To solve this issue, the model-based deep learning methods build the network by unrolling the specific optimization algorithm [23, 24]. However, the optimization algorithms of model-based deep learning methods still require well-designed priors or assumptions. Additionally, the convergence of the optimization algorithms is not taken into account in the design of the unrolling networks.

To address these problems, we propose a novel pan-sharpening approach called *alternating reverse filtering network*, which combines classical reverse filtering [25] and deep learning. Unlike previous methods, we formulate pan-sharpening as a reverse filtering process, thus avoiding the dependency on pre-defined priors or assumption. In addition, we tailor the classical reverse filtering in an alternating iteration manner for the pan-sharpening problem. The process of alternating iterations is unrolled into a network. We demonstrate such formulations is theoretical feasibility. To guarantee the stability and convergence of the iterative process on the basis of contraction mapping in a metric space, we introduce the learnable multi-scale Gaussian kernel module in the network. Such a key issue is commonly neglected in previous unrolling-based deep learning methods.

The main contributions of this paper can be summarized as follows: 1) We introduce a new perspective for pan-sharpening by formulating it as a reverse filtering process. To the best of our knowledge, this is the first effort to solve multispectral image fusion problem using the method of this kind. 2) In contrast to existing model-driven methods, our iterative network can obtain HR MS without the need for pre-defined exact priors or assumptions. 3) Instead of using specific filters in reverse filtering, we constrain their formulation in a more compact learnable multi-scale Gaussian kernel module, which guarantees the stability and convergence of the iterative process. In addition, extensive experimental results on simulated and real-world scenes show that the proposed network significantly outperforms the state of the arts.

## 2 Related Work

**Pan-sharpening.** Traditional pan-sharpening methods are classified into three types: component substitution (CS)-, multi-resolution analysis (MRA)-, and variational optimization (VO)-based methods [26, 16, 27–29]. The main idea of CS-based methods is to separate spatial and spectral information of the MS image in a suitable space and further fuse them with the PAN image. The representative CS-based methods include intensity hue-saturation (IHS) fusion [30], the principal component analysis (PCA) methods [11, 31], Brovey transforms [32], and Gram-Schmidt (GS) orthogonalization method [33]. MRA-based methods decompose MS and PAN images into multi-scale space via decimated wavelet transform (DWT) [34], high-pass filter fusion (HPF) [35], indusion method [13] and atrous wavelet transform (ATWT) [36]. Then, the decomposed version PAN images are injected into the corresponding MS images for information fusion. VO-based methods regard the pan-sharpening tasks as an ill-posed problem by minimizing a loss function, including dynamic gradient sparsity property (SIRF) [37], local gradient constraint (LGC) [15], group low-rank constraint for texture similarity (ADMM) [16]. However, the performance of these methods is limited due to the shallow non-linear expression in these models. Since then, deep learning-based pan-sharpening algorithms have dominated this field [9, 38, 39]. Masi *et al.* [9] are the first to use CNN to deal with the issue of pan-sharpening. Although the structure is simple, the effect is much better than the traditional methods. Then, Yang *et al.* [38] designed a deeper CNN by relying on resblock in [40]. Meanwhile, Yuan *et al.* [41] introduced multi-scale module into the basic CNN architecture.

**Unrolling-based deep learning method.** In recent years, many researchers [42–47] attempt to combine domain knowledge with deep neural networks to propose deep unrolling networks which take advantages of the model-based methods' interpretability and learning-based methods' strong mapping ability. Specifically, the deep unrolling network firstly unrolls certain optimization algorithms [48–50, 23, 51–57] and utilizes deep neural network to parameterize the unrolling model, then minimizes the loss function and optimizes the parameters in an end-to-end manner. For example, Xu *et al.* [24] developed two separate priors of PAN and MS modality to design the unrolling structure for

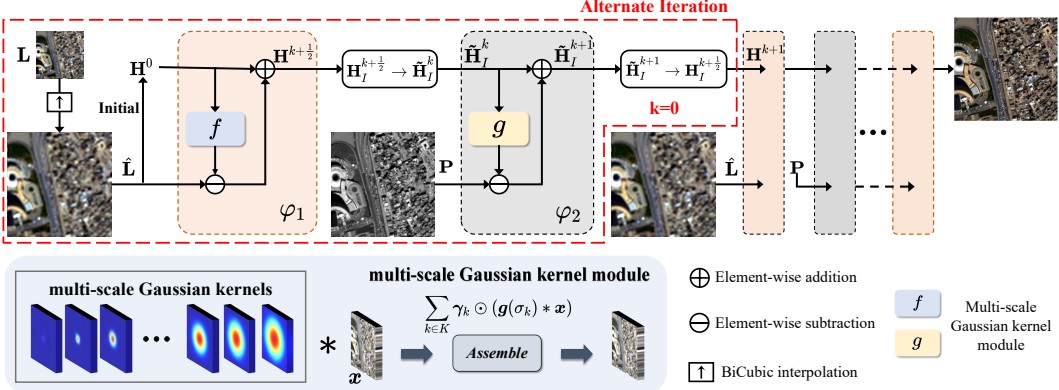

Figure 1: The overall architecture of alternating reverse filtering network.

pan-sharpening. The model-driven methods have better interpretability and clearer physical meaning. Cao *et al.* [46] unrolled an alternate optimization algorithm into CNN. However, the optimization of model-based deep methods also require well-designed priors and exact degradation assumptions.

## 3 Proposed Method

In this section, we provide a detailed introduction to our proposed alternating reverse filtering network for pan-sharpening. For convenience, we first define some notations. Concretely, $\mathbf{L} \in \mathbb{R}^{w \times h \times B}$ denotes the low-resolution (LR) multispectral image, $\mathbf{H} \in \mathbb{R}^{W \times H \times B}$ represents the corresponding high-resolution (HR) multispectral image, and $\mathbf{P} \in \mathbb{R}^{W \times H \times b}$ is the PAN image.

### 3.1 Problem Formulation

For multispectral image restoration, the degradation model is commonly formulated as

$$\mathbf{L} = (\mathbf{H} * \mathbf{k}) \downarrow_{\mathbf{s}} + \boldsymbol{\epsilon}, \tag{1}$$

where $*$, $\mathbf{k}$, $\downarrow_{\mathbf{s}}$ and $\boldsymbol{\epsilon}$ denote the convolution operation, blurring kernel, down-sampling operator and the measurement noise, respectively. To restore high quality $\mathbf{H}$ from $\mathbf{L}$, high-resolution PAN image $\mathbf{P}$ is introduced to help enhance structural detail. In CS-based models, the intensity component $\mathbf{I}$, a generalized IHS concept [58], is usually replaced by $\mathbf{P}$ but the discrepancy between $\mathbf{P}$ and $\mathbf{I}$ can cause spectral distortion in the fused image. The VO-based models solve the problem by explicitly constructing many well-designed priors among $\mathbf{H}$, $\mathbf{L}$ and $\mathbf{P}$. However, hand-crafted priors don't work well in practical complex situations. Inspired by classical reverse filtering [25], we propose alternating reverse filtering method to estimate $\mathbf{H}$ by the more general multispectral image priors:

$$\mathbf{L} = f(\mathbf{H}), \tag{2}$$
$$\mathbf{P} = g(\mathbf{H}_I), \tag{3}$$

where $f(\cdot)$ and $g(\cdot)$ denote the degradation processes, and $\mathbf{H}_I$ is the intensity component of $\mathbf{H}$.

### 3.2 Model Optimization

**Definition 3.2** *Suppose $(\mathcal{H}, d)$ is a metric space and $T : \mathcal{H} \to \mathcal{H}$ is a mapping function. For all $x, y \in \mathcal{H}$, if there exists a constant $c \in [0, 1)$ that makes the following formula*

$$d(T(x), T(y)) \leq cd(x, y), \tag{4}$$

*mapping $T : \mathcal{H} \to \mathcal{H}$ is called Contraction Mapping*[59].

In the image space, the metric space $(\mathcal{H}, d)$ can be expressed as

$$\mathcal{H} = R^{w \times h}, d(x, y) = \|x - y\|, \tag{5}$$

where $w \times h$ is the number of image pixels and $d(x, y)$ is the Euclidean distance.

**Theorem 3.2** *A variable $x^*$ is a fixed point for a given function $\Phi$ if $\Phi(x^*) = x^*$. When mapping $\Phi : \mathcal{H} \to \mathcal{H}$ is a contraction mapping, $\Phi$ admits a unique fixed-point $x^*$ in $\mathcal{H}$. Further, $x^*$ can be found in the following way. Let the initial guess be $x_0$ and define a sequence $\{x_n\}$ as $x_n = \Phi(x_{n-1})$. When the iterative process converges, $\lim_{n \to \infty} x_n = x^*$.*

**Reverse Filtering** Without loss of generality, the degenerate processes $f(\cdot)$ and $g(\cdot)$ can be considered as broadly defined filters $F(\cdot)$ that smooth images, texture removal or other properties. Filtering process can be described as

$$\boldsymbol{y} = F(\boldsymbol{x}), \tag{6}$$

where $\boldsymbol{x}$ and $\boldsymbol{y}$ are the input image and the filtering result. When $F(\cdot)$ is unknown, it's difficult to apply well-designed image priors to obtain the $\boldsymbol{x}$. However, reverse filtering can estimate $\boldsymbol{x}$ without needing to compute $F^{-1}(\cdot)$ and update restored images according to the filtering effect as

$$\boldsymbol{x}^{k+1} = \boldsymbol{x}^k + \boldsymbol{y} - F(\boldsymbol{x}^k), \tag{7}$$

where $\boldsymbol{x}^k$ is the current estimate of $\boldsymbol{x}$ in the $k$-th iteration. The iteration starts from $\boldsymbol{x}^0 = \boldsymbol{y}$ and $\boldsymbol{x}^k$ gets closer and closer to $\boldsymbol{x}$ with the increasing $k$. We make auxiliary function $\varphi(\cdot)$ as

$$\varphi(\boldsymbol{x}) = \boldsymbol{x} + \boldsymbol{y} - F(\boldsymbol{x}). \tag{8}$$

Therefore, the above iterative process can be regarded as a fixed point iteration

$$\boldsymbol{x}^{k+1} = \varphi(\boldsymbol{x}^k). \tag{9}$$

With the above analysis, we take the filtering function $F(\cdot)$ in Equation 8 as $f(\cdot)$ and $g(\cdot)$ in Equation 2 to obtain two reverse filtering:

$$\begin{cases} \varphi_1(\mathbf{H}) & = & \mathbf{H} + \hat{\mathbf{L}} - f(\mathbf{H}) \\ \varphi_2(\mathbf{H}_I) & = & \mathbf{H}_I + \mathbf{P} - g(\mathbf{H}_I), \end{cases} \tag{10}$$

and hence our alternating reverse filtering method can be written by following

$$\begin{cases} \mathbf{H}^{k+\frac{1}{2}} & = & \mathbf{H}^k + \hat{\mathbf{L}} - f(\mathbf{H}^k) \\ \tilde{\mathbf{H}}_I^k & = & \mathbf{H}_I^{k+\frac{1}{2}} \\ \tilde{\mathbf{H}}_I^{k+1} & = & \tilde{\mathbf{H}}_I^k + \mathbf{P} - g(\tilde{\mathbf{H}}_I^k) \\ \mathbf{H}^{k+1} & \Leftarrow & (\mathbf{H}_I^{k+\frac{1}{2}} \leftarrow \tilde{\mathbf{H}}_I^{k+1}), \end{cases} \tag{11}$$

where $\hat{\mathbf{L}}$ is the upsampled LR multispectral image $\mathbf{L}$ and $\tilde{\mathbf{H}}_I^k$ is an approximate estimate of $\mathbf{H}_I$. Equation 11 starts with initial state $\mathbf{H}^0 = \hat{\mathbf{L}}$. Note that the calculated $\mathbf{H}^{k+\frac{1}{2}}$ is used as input in the next iteration, the intensity component $\mathbf{H}_I^{k+\frac{1}{2}}$ is then sent into another iteration $\tilde{\mathbf{H}}_I^{k+1} = \tilde{\mathbf{H}}_I^k + \mathbf{P} - g(\tilde{\mathbf{H}}_I^k)$ to enhance the structural details with the help of $\mathbf{P}$ image. After that, the enhanced intensity component replaces the original component $\mathbf{H}_I^{k+\frac{1}{2}} \leftarrow \tilde{\mathbf{H}}_I^{k+1}$ and the new $\mathbf{H}^{k+1}$ is used as input in the next iteration.

## 3.3 Alternating Reverse Filtering Network

The end-to-end model we construct for GMIR, named as ARFNet (Alternating Reverse Filtering Network), is based on two fixed point iterations in Equation 11 with multi-scale Gaussian convolution module acting as filters. See Figure 1 for the overview of the proposed method. If reverse filtering satisfies the sufficient condition for definition 3.2, ARFNet will converge $\lim_{k \to \infty} \mathbf{H}^k = \mathbf{H}^*$ and finally reaches $f(\mathbf{H}^*) \approx \hat{\mathbf{L}}$. Take the $\varphi_1(\cdot)$ for instance, the sufficient condition that theorem 3.2 holds is that $\varphi_1(\mathbf{H})$ forms a contraction mapping

$$\begin{aligned} \|\varphi_1(\mathbf{H}_a) - \varphi_1(\mathbf{H}_b)\| &= \left\| \left[ \mathbf{H}_a + \hat{\mathbf{L}} - f(\mathbf{H}_a) \right] - \left[ \mathbf{H}_b + \hat{\mathbf{L}} - f(\mathbf{H}_b) \right] \right\| \\ &= \|[\mathbf{H}_a - f(\mathbf{H}_a)] - [\mathbf{H}_b - f(\mathbf{H}_b)]\| \le c \cdot \|\mathbf{H}_a - \mathbf{H}_b\|, \quad c \in [0, 1) \end{aligned} \tag{12}$$

For linear filters, the condition is further simplified as

$$\|\mathbf{H} - f(\mathbf{H})\| \le c \cdot \|\mathbf{H}\|. \quad c \in [0, 1) \tag{13}$$

In our case, the degenerate processes $f(\cdot)$ and $g(\cdot)$ are implemented by multi-scale Gaussian convolution modules.

**Multi-scale Gaussian Convolution Module**    Given an input image $\boldsymbol{x}$, the output $\boldsymbol{y}$ of Gaussian convolution [60] can be expressed as

$$\boldsymbol{y} = \boldsymbol{g}(\sigma_k) * \boldsymbol{x}, \tag{14}$$

where $\sigma_k$ denotes the variance of 2D Gaussian kernels. In particular, we define $\sigma_1 = 0$ and $\boldsymbol{g}(\sigma_1) = \boldsymbol{\delta}$ which is Dirac delta. Therefore,

$$\|\mathbf{H} * (\boldsymbol{g}(\sigma_1) - \boldsymbol{g}(\sigma))\| = \|\widehat{\mathbf{H}} \odot (\mathbf{1} - \widehat{\boldsymbol{g}}(\boldsymbol{\sigma}))\| \leq c \cdot \|\widehat{\mathbf{H}}\| = c \cdot \|\mathbf{H}\|, \tag{15}$$

where $\widehat{H}$ denotes Fourier transform, $\odot$ denotes point-wised product and $\mathbf{1}$ denotes all-one matrix. Thus, the inequality holds true when $\boldsymbol{g}(\sigma_k)$ is a normalized Gaussian kernel that means these filters can be strictly reversed using fixed-point iteration. In our model, normalized Gaussian kernel is used for initialization and its parameters can be further learned from the data in an end-to-end manner. Although the learned convolution kernels may not completely satisfy the condition, fixed point iteration can be split into two independent sequences:

$$\mathbf{H}^{k+1} = \mathbf{H}_c^{k+1} + \mathbf{H}_o^{k+1} = \varphi_1\left(\mathbf{H}_c^k\right) + \varphi_1\left(\mathbf{H}_o^k\right), \tag{16}$$

where $\{\mathbf{H}_c^k\}$ is guaranteed to converge to the unique solution and $\{\mathbf{H}_o^k\}$ could oscillate. Fortunately, after the first few epochs, the learned convolution kernels are close to the kernels we initialize, which makes $\{\mathbf{H}_c^k\}$ the majority and dominates the convergence of the whole process. In the implementation of the algorithm, we adopt multi-scale Gaussian convolution module to obtain better filtering effect. Compared to Gaussian convolution, the multi-scale Gaussian convolution module integrates Gaussian kernels with different kernel sizes, expressed as

$$\boldsymbol{y} = \sum_{k \in K} \boldsymbol{\gamma}_k \odot (\boldsymbol{g}(\sigma_k) * \boldsymbol{x}), K = \{1, 3, 5, \cdots, M\}, \tag{17}$$

where $\gamma_k$ is the learnable mixing coefficient and $k$ denotes different kernel sizes $1 \times 1, 3 \times 3, 5 \times 5, \cdots, M \times M$. The multi-scale Gaussian convolution is a series simple group convolutional layers defined by initialized 2D Gaussian kernels. Clearly, the form of weighted summation still conforms to the above analysis about sufficient condition.

Taken together, the forward process of alternating reverse filtering network can be described as Algorithm 1.

---

**Algorithm 1** Proposed algorithm.

---

**Input:** The upsampled low-resolution multispectral image $\hat{\mathbf{L}}$, panchromatic image $\mathbf{P}$ and maximum iteration number $K$.

    initial $\mathbf{H}^0 = \hat{\mathbf{L}}$;

    **for** $k = 0, 1, 2, 3, \cdots, K$ **do**

        compute $\mathbf{H}^{k+\frac{1}{2}} = \mathbf{H}^k + \hat{\mathbf{L}} - f(\mathbf{H}^k)$;

        fetch the intensity component $\tilde{\mathbf{H}}_I^k = \mathbf{H}_I^{k+\frac{1}{2}}$;

        compute $\tilde{\mathbf{H}}_I^{k+1} = \tilde{\mathbf{H}}_I^k + \mathbf{P} - g(\tilde{\mathbf{H}}_I^k)$;

        replace the intensity component $\mathbf{H}_I^{k+\frac{1}{2}} \leftarrow \tilde{\mathbf{H}}_I^{k+1}$ to get $\mathbf{H}^{k+1}$;

    **end for**

**Output:** fused high-resolution multispectral image $\mathbf{H}^K$ and estimated intensity component $\tilde{\mathbf{H}}_I^K$.

---

**Loss Function**    We utilize two loss functions, there are the reconstruction loss $\mathcal{L}_r$ and the structure loss $\mathcal{L}_s$, as following

$$\mathcal{L}_{\text{sum}} = \mathcal{L}_{\text{r}} + \lambda \mathcal{L}_{\text{s}}, \tag{18}$$

where $\lambda$ is the hyperparameter which determines the balance between the overall performance and the structure texture details. Specifically, reconstruction loss $\mathcal{L}_r$ is a common pixel-wise L2 loss and structure loss $\mathcal{L}_s$ is based on structural similarity (SSIM). Thus, the corresponding losses are defined as follows:

$$\mathcal{L}_{\text{r}} = \|\mathbf{H}^K - \mathbf{H}\|_2, \tag{19}$$

$$\mathcal{L}_{\text{s}} = 1 - SSIM(\tilde{\mathbf{H}}_I^K, \mathbf{H}_I), \tag{20}$$

where $\mathbf{H}$, $\mathbf{H}_I$, $\mathbf{H}^K$ and $\tilde{\mathbf{H}}_I^K$ are the ground truth, intensity component of $\mathbf{H}$, the output of alternating reverse filtering network $\varphi_1(\cdot)$ and $\varphi_2(\cdot)$ respectively.

Table 1: Quantitative comparison with the state-of-the-art methods. The best results are highlighted by **bold**. The ↑ or ↓ indicates higher or lower values correspond to better results.

| Method | WordView II | | | | GaoFen2 | | | | WordView III | | | |
|---|---|---|---|---|---|---|---|---|---|---|---|---|
| | PSNR↑ | SSIM↑ | SAM↓ | ERGAS↓ | PSNR↑ | SSIM↑ | SAM↓ | ERGAS↓ | PSNR↑ | SSIM↑ | SAM↓ | ERGAS↓ |
| SFIM | 34.1297 | 0.8975 | 0.0439 | 2.3449 | 36.9060 | 0.8882 | 0.0318 | 1.7398 | 21.8212 | 0.5457 | 0.1208 | 8.9730 |
| Brovey | 35.8646 | 0.9216 | 0.0403 | 1.8238 | 37.7974 | 0.9026 | 0.0218 | 1.3720 | 22.506 | 0.5466 | 0.1159 | 8.2331 |
| GS | 35.6376 | 0.9176 | 0.0423 | 1.8774 | 37.2260 | 0.9034 | 0.0309 | 1.6736 | 22.5608 | 0.5470 | 0.1217 | 8.2433 |
| IHS | 35.2962 | 0.9027 | 0.0461 | 2.0278 | 38.1754 | 0.9100 | 0.0243 | 1.5336 | 22.5579 | 0.5354 | 0.1266 | 8.3616 |
| GFPCA | 34.5581 | 0.9038 | 0.0488 | 2.1411 | 37.9443 | 0.9204 | 0.0314 | 1.5604 | 22.3344 | 0.4826 | 0.1294 | 8.3964 |
| PNN | 40.7550 | 0.9624 | 0.0259 | 1.0646 | 43.1208 | 0.9704 | 0.0172 | 0.8528 | 29.9418 | 0.9121 | 0.0824 | 3.3206 |
| PANNet | 40.8176 | 0.9626 | 0.0257 | 1.0557 | 43.0659 | 0.9685 | 0.0178 | 0.8577 | 29.684 | 0.9072 | 0.0851 | 3.4263 |
| MSDCNN | 41.3355 | 0.9664 | 0.0242 | 0.9940 | 45.6874 | 0.9827 | 0.0135 | 0.6389 | 30.3038 | 0.9184 | 0.0782 | 3.1884 |
| SRPPNN | 41.4538 | 0.9679 | 0.0233 | 0.9899 | 47.1998 | 0.9877 | 0.0106 | 0.5586 | 30.4346 | 0.9202 | 0.0770 | 3.1553 |
| GPPNN | 41.1622 | 0.9684 | 0.0244 | 1.0315 | 44.2145 | 0.9815 | 0.0137 | 0.7361 | 30.1785 | 0.9175 | 0.0776 | 3.2596 |
| Ours | **41.7587** | **0.9691** | **0.0229** | **0.9540** | **47.2238** | **0.9892** | **0.0102** | **0.5495** | **30.5425** | **0.9216** | **0.0768** | **3.1049** |

## 4 Experiments

In this section, the datasets and experimental settings are firstly described. Then, we evaluate the effectiveness of our proposed alternating reverse filtering network (ARFNet) on simulated and real-world full-resolution scenes. Additionally, we conduct an ablation study to gain insight into the respective effect of different parameter configurations. More experimental results are included in the supplemental material.

### 4.1 Datasets and Experimental Settings

In order to verify the effectiveness of our models for GMIR, multispectral and panchromatic images obtained on three commercial satellites that widely are used, including WorldViewII (WV2), WorldViewIII (WV3), and GaoFen2 (GF2). Each database contains thousands of image pairs, and they are divided into training, validation and testing sets that follow the prior works to generate the training set by employing the Wald protocol tool [61]. In the training set, each training pair contains one guided PAN image with the size of $128 \times 128$, one LR MS patch with the size of $32 \times 32$, and one ground truth HR MS patch with the size of $128 \times 128$.

Models are implemented via PyTorch and one NVIDIA GTX 3090 GPU is used for training. In the experiments, the SGD algorithm with a momentum equals to 0.9 is adopted to train the models and the minibatch size is set to 4. The initial learning rate is set to $1 \times 10^{-2}$. When reaching 1000 and 1500 epochs, the learning rate is decayed by multiplying 0.5, and training ends after 2000 epochs. Through all experiments, We set the hyperparameter $\lambda$ in loss function 18 to 0.1, the number $K$ of alternate iteration to 5 and the maximum size $M$ of Gaussian kernels to 17. The sigma of the Gaussian function is set to one fourth of the kernel size.

### 4.2 Comparison with SOTAs

We conduct several experiments on the benchmark datasets compared with several representative guided multispectral image restoration methods: five promising traditional methods, including smoothing filter-based intensity modulation ((SFIM) [62], Brovey [32], GS [33], intensity hue-saturation fusion (IHS) [63], and PCA guided filter (GFPCA) [64]; five commonly-recognized state-of-the-art deep-learning based methods, including PNN [9], PANNET [38], multiscale and multidepth network (MSDCNN) [41], super-resolution-guided progressive network (SRPPNN) [65], and deep gradient projection network (GPPNN) [66].

In our experiments, we select the widely-used image quality assessment (IQA) metrics for evaluation such as the peak signal-to-noise ratio (PSNR), the structural similarity (SSIM), the relative dimension-

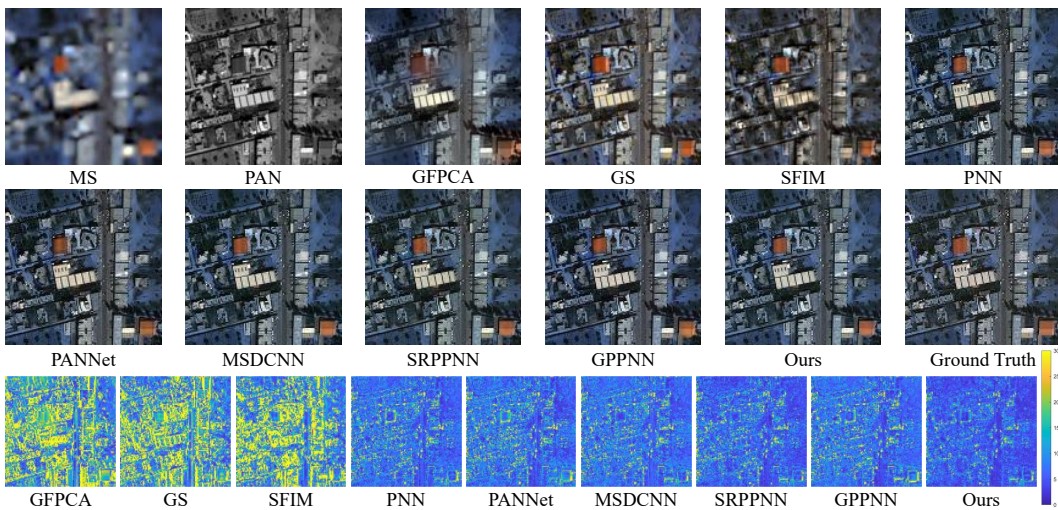

Figure 2: Qualitative visualization comparison of our method with other representative counterparts on a typical satellite image pair from the WordView-III dataset. Images in the last row visualizes the MSE between the fused images and the ground truth.

Table 2: The average quantitative results on the GaoFen2 datasets on the real-world full-resolution scene. The best results are highlighted by **bold**.

| Metrics | SFIM | GS | Brovey | IHS | GFPCA | PNN | PANNET | MSDCNN | SRPPNN | GPPNN | **Ours** |
|---|---|---|---|---|---|---|---|---|---|---|---|
| $D_\lambda \downarrow$ | 0.0822 | 0.0696 | 0.1378 | 0.0770 | 0.0914 | 0.0746 | 0.0737 | 0.0734 | 0.0767 | 0.0782 | **0.0635** |
| $D_s \downarrow$ | **0.1087** | 0.2456 | 0.2605 | 0.2985 | 0.1635 | 0.1164 | 0.1224 | 0.1151 | 0.1162 | 0.1253 | 0.1156 |
| $QNR \uparrow$ | 0.8214 | 0.7025 | 0.6390 | 0.6485 | 0.7615 | 0.8191 | 0.8143 | 0.8215 | 0.8173 | 0.8073 | **0.8237** |

less global error in synthesis (ERGAS) [67], the correlation coefficient (SCC), the four-band extension of Q, the spectral angle mapper (SAM) [68], the spectral distortion index $D_\lambda$, the spatial distortion index $D_S$, the quality without reference (QNR) [69]. The last three indicators are non-reference metrics.

**Results on Simulated Scene** To quantitatively compare the fused multispectral images with the paired reference ground truth images offered on the simulated datasets, we conduct repeated experiments on three datasets. The average performance of representative GMIR methods is tabled in Table 1. The higher values of PSNR and SSIM, the more similar structure between fused multispectral images and ground truth images. ERGAS takes into account the relative errors of all channels. SAM, Q and SCC focus on measuring spectral distortion. More experimental metric results are included in the supplemental material. The qualitative comparison of the visual results over the representative sample from the WorldView-III dataset is in Figure 2. To highlight the differences in detail, we show the error map between fused image and ground truth image in the last row. Similarly, more qualitative comparisons are shown in the supplemental material.

**Results on Real-world Full-resolution Scene** To assess the generalization performance of models in real-world scene, we apply models trained on the GF2 dataset to additional 200 paired GaoFen2 satellite images which are constructed using the full-resolution setting as the real scene. Lacking available ground-truth at full-resolution scene, we employ three widely-used non-reference metrics for assessing the performance: $D_\lambda$, $D_S$ and QNR. The quantitative and qualitative results are summarized in Table 2 and Figure 3 that clearly demonstrate the higher generalization capacity of the proposed alternating reverse filtering network.

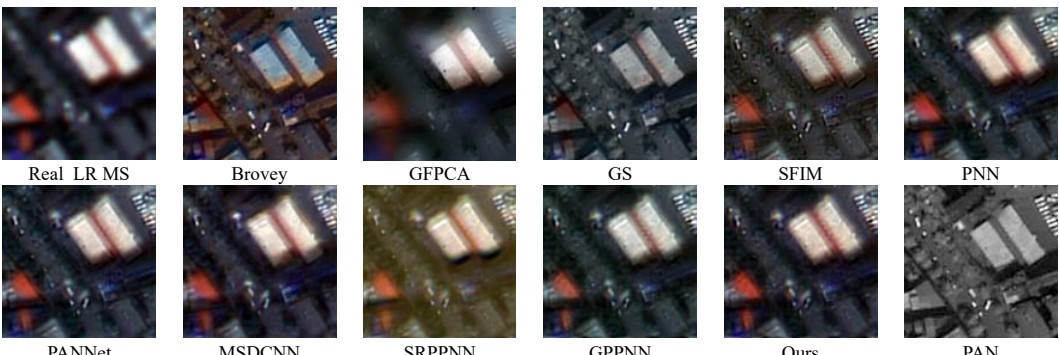

Figure 3: Qualitative visualization comparison of our method with other representative methods over real-world full-resolution scenes.

Table 3: The effect of hyperparameter $\lambda$ in loss function $\mathcal{L}_{sum}$.

| $\lambda$ | 0 | 0.001 | 0.01 | 0.1 | 0.5 | 1 | 2 |
|---|---|---|---|---|---|---|---|
| PSNR↑ | 40.1274 | 40.8279 | 41.3267 | **41.7587** | 41.6932 | 41.2736 | 41.0746 |
| SSIM↑ | 0.9598 | 0.9630 | 0.9673 | 0.9691 | **0.9694** | 0.9664 | 0.9653 |
| SAM↓ | 0.0264 | 0.0256 | 0.0242 | **0.0229** | 0.0232 | 0.247 | 0.0260 |
| ERGAS↓ | 1.0677 | 1.0561 | 0.9943 | **0.9540** | 0.9564 | 1.0037 | 1.0491 |

Table 4: Quantitative comparison of different initialization methods on the WorldView-II dataset.

| Methods | PSNR↑ | SSIM↑ | SAM↓ | ERGAS↓ | SCC↑ | Q↑ | $D_\lambda \downarrow$ | $D_S \downarrow$ | QNR↑ |
|---|---|---|---|---|---|---|---|---|---|
| (I) | 40.3297 | 0.9601 | 0.0262 | 1.0672 | 0.9663 | 0.7310 | 0.0698 | 0.1277 | 0.8097 |
| (II) | **41.7587** | **0.9691** | **0.0229** | **0.9540** | **0.9749** | **0.7731** | **0.0631** | **0.1184** | **0.8285** |
| (III) | 40.4161 | 0.9612 | 0.0261 | 1.0668 | 0.9667 | 0.7316 | 0.0684 | 0.1275 | 0.8123 |

## 4.3 Ablation Study

Ablation studies are implemented on the WordView-II dataset to explore the effect of different parameters and components on the performance of models. We use the ARFNet in subsection 4.1 as the baseline for comparison by changing parameters and components, and all comparison models are trained in the same way. Firstly, to balance the overall performance and the structure details, we fix the other components and change only the value of the hyperparameter $\lambda$. The results in Table 3 show that the larger the value of $\lambda$ is within a certain range, the higher the structure similarity between the fused image and ground truth will be. Note that if the reverse filtering network $\varphi_2(\cdot)$ lack the supervision $\mathcal{L}_s$ that the performance of the entire network will be a significant decrease.

Furthermore, we compare the results of different initialization methods. In Table 4, (I) represents the kernels that are randomly initialized by Kaiming [70], (II) and (III) represents the kernels that are initialized by Gaussian kernels, but the kernels in (III) are fixed. From Table 4 and Figure 4, one could see that: 1) The kernels learned by the randomly initialized network cannot satisfy the sufficient condition, which leads to poor performance. 2) The learned kernels initialized by Gaussian kernels are close to the kernels we initialize, that makes $\{\mathbf{H}_c^k\}$ the majority and dominates the convergence of the whole process. 3) Although the learned kernels are close to the initialized Gaussian kernels, a large number of multi-scale Gaussian kernels can still bring performance improvements.

As can be seen from Table 5, the ablation studies about maximum kernel size $M$ show that the multi-scale Gaussian convolution module can bring better performance improvement. To explore the impact of the number of iterations $K$ on the performance, we experiment with varying numbers of $K$. Table 6 shows the results of different $K$ from 1 to 7. It can be seen that the PSNR performance

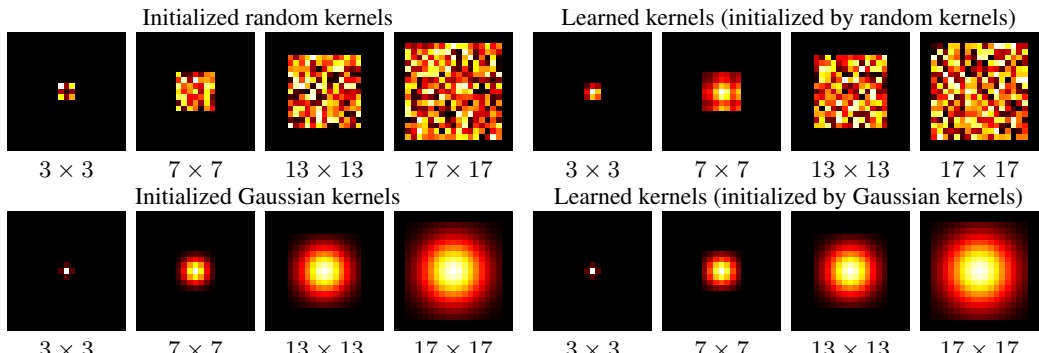

Figure 4: Visualization of initialized kernels and learned kernels.

increases as the number of stages increases. We choose $K = 5$ in our implementation to balance the performance and computational complexity.

Table 5: The comparison results of ablation study about maximum kernel size $M$.

| $M$ | 1 | 3 | 5 | 7 | 9 | 11 | 13 | 15 | 17 | 19 | 21 |
|---|---|---|---|---|---|---|---|---|---|---|---|
| PSNR↑ | 39.7820 | 40.6804 | 41.0634 | 41.4371 | 41.5297 | 41.6445 | 41.6910 | 41.7252 | **41.7587** | 41.7126 | 41.6982 |
| SSIM↑ | 0.9584 | 0.9603 | 0.9657 | 0.9665 | 0.9679 | 0.9684 | 0.9688 | 0.9690 | **0.9691** | 0.9689 | 0.9683 |

### 4.4 Limitations and Discussions

First, we evaluate the effectiveness of the proposed framework over panchromatic and multispectral image fusion and we will extend the framework to other multispectral fusion tasks, such as RGB and multispectral image fusion. Second, although we develop the multi-scale Gaussian kernel module to ensure the convergence of the alternating reverse filtering, there is still a large room to explore a learnable filter function that can strictly satisfy sufficient conditions.

## 5 Conclusion

In this paper, we presented a simple yet effective alternating reverse filtering network for pan-sharpening. The proposed approach formulates pan-sharpening as a reverse filtering process and combines classical reverse filtering and deep learning. The classical reverse filtering is unrolled to a network without pre-defined exact priors in an alternating iteration manner. Besides, multi-scale Gaussian kernel module is developed to ensure the convergence of the iterative process. Furthermore, the ablation studies verified the effectiveness of the multi-scale Gaussian kernel module. Extensive experimental results on simulated and real-world scenes show that the proposed network significantly outperforms the state of the arts. In the future, we will study the extension to other image fusion problems such as RGB and multispectral image fusion.

## Broader Impact

This research aims to address the problem of panchromatic and multispectral image fusion, which is a key pre-processing technology overcoming the constraints of hardware before using high-resolution multispectral image. The fused multispectral images are needed as a reference in the field of resource monitoring, such as land use planning, ocean development, and urban management, and in the field of ecological and environmental protection areas, such as pollution monitoring, vegetation biology and precision agriculture research. Despite the many benefits of fused multispectral images, negative consequences can still occur in several special environments. When there is a case of algorithm failure, artifacts generated on the fusion image may affect subsequent use and lead to misjudgments.

Table 6: The comparison results of ablation study about the number of iterations $K$.

| $K$ | 1 | 2 | 3 | 4 | 5 | 6 | 7 |
|---|---|---|---|---|---|---|---|
| PSNR↑ | 40.7130 | 41.056 | 41.3927 | 41.6869 | 41.7587 | **41.7614** | 41.7603 |
| SSIM↑ | 0.9611 | 0.9654 | 0.9667 | 0.9684 | **0.9691** | 0.9690 | 0.9688 |

## Acknowledgements

This work was supported in part by the National Natural Science Foundation of China under Grant 42271350 and the University Synergy Innovation Program of Anhui Province under Grant GXXT-2019-025. We gratefully acknowledge the support of MindSpore, CANN, and Ascend AI Processor used for this research.

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
