# Panchromatic and Multispectral Image Fusion via Alternating Reverse Filtering Network (Supplementary Materials)

**Keyu Yan**[1,2][*]**, Man Zhou**[1,2][*]**, Jie Huang**[2]**, Feng Zhao**[2]**, Chengjun Xie**[1]**, Chongyi Li**[3]**,
Danfeng Hong**[4][†]
[1]Hefei Institute of Physical Science, Chinese Academy of Sciences, China
[2]University of Science and Technology of China, China
[3]Nanyang Technological University, Singapore
[4]Aerospace Information Research Institute, Chinese Academy of Sciences, China

## 1 Experimental Results

### 1.1 Quantitative comparison

We show the quantitative experiment comparisons of several measurement metrics, including the peak signal-to-noise ratio (PSNR), the structural similarity (SSIM), the spectral angle mapper (SAM), the relative dimensionless global error in synthesis (ERGAS), the correlation coefficient (SCC), Q-index, the three non-reference metrics of the spectral distortion index ($D_\lambda$), the spatial distortion index ($D_S$) and the quality without reference (QNR) in Table 1 and Table 2, Table 3 between our predictions and that of the baseline methods on WordView-II, GaoFen2 and WordView-III dataset, respectively. The best results are highlighted by bold. It can be clearly seen that our alternating reverse filtering network performs the best compared with other state-of-the-art methods in all the indexes, indicating the superiority of our proposed method.

### 1.2 Qualitative comparison

The qualitative visualization comparisons between our method and several state-of-the-art pan-sharpening methods are shown in Figure 1 and Figure 2 on the representative samples of the WordView-II and GaoFen2 datasets. Images in the last row are the MSE residues between the fused results and the ground truth. Compared with other competing methods, our model has minor spatial and spectral distortions. It can be easily concluded from the observation of MSE maps. As for the MSE residues, it's noticed that our proposed method is closest to the ground truth than other comparison methods. Therefore, It can be affirmed that our method achieves the best performance than other competitive pan-sharpening algorithms.

## 2 Visual Examples of Intermediate Results

The iterative procedure progressively recovers the structure details of the fused image according to the filtering effect. The intermediate visual results of alternating reverse filtering network with different iterations are shown in Figure 3, from which we can observe that more detailed information is recovered along with the greater number of iterations.

---

[*]Co-first authors contributed equally, [†] corresponding author.

36th Conference on Neural Information Processing Systems (NeurIPS 2022).

Table 1: Quantitative comparison of nine metrics on the WordView-II dataset. Best results are highlighted by **bold**. ↑ indicates that the larger the value, the better the performance, and ↓ indicates that the smaller the value, the better the performance.

| Methods | PSNR ↑ | SSIM ↑ | SAM ↓ | ERGAS ↓ | SCC ↑ | Q ↑ | $D_\lambda$ ↓ | $D_S$ ↓ | QNR ↑ |
|---|---|---|---|---|---|---|---|---|---|
| SFIM | 34.1297 | 0.8975 | 0.0439 | 2.3449 | 0.9079 | 0.6064 | 0.0915 | 0.1277 | 0.7942 |
| GS | 35.6376 | 0.9176 | 0.0423 | 1.8774 | 0.9225 | 0.6307 | 0.0607 | 0.1285 | 0.8195 |
| Brovey | 35.8646 | 0.9216 | 0.0403 | 1.8238 | 0.8913 | 0.6163 | 0.077 | 0.136 | 0.7977 |
| IHS | 35.2962 | 0.9027 | 0.0461 | 2.0278 | 0.8534 | 0.5704 | 0.0774 | 0.1578 | 0.777 |
| GFPCA | 34.5581 | 0.9038 | 0.0488 | 2.1411 | 0.8924 | 0.4665 | 0.1016 | 0.1656 | 0.7508 |
| PNN | 40.7550 | 0.9624 | 0.0259 | 1.0646 | 0.9677 | 0.7426 | 0.065 | 0.1186 | 0.825 |
| PANNet | 40.8176 | 0.9626 | 0.0257 | 1.0557 | 0.968 | 0.7437 | 0.0645 | 0.1189 | 0.8252 |
| MSDCNN | 41.3355 | 0.9664 | 0.0242 | 0.9940 | 0.9721 | 0.7577 | 0.0635 | 0.1172 | 0.8276 |
| SRPPNN | 41.4538 | 0.9679 | 0.0233 | 0.9899 | 0.9729 | 0.7691 | 0.0637 | 0.1164 | 0.8281 |
| GPPNN | 41.1622 | 0.9684 | 0.0244 | 1.0315 | 0.9722 | 0.7627 | 0.0642 | 0.1163 | 0.8278 |
| Ours | **41.7587** | **0.9691** | **0.0229** | **0.9540** | **0.9749** | **0.7731** | **0.0631** | **0.1184** | **0.8285** |

Table 2: Quantitative comparison of nine metrics on the GaoFen2 dataset. Best results are highlighted by **bold**. ↑ indicates that the larger the value, the better the performance, and ↓ indicates that the smaller the value, the better the performance.

| Methods | PSNR ↑ | SSIM ↑ | SAM ↓ | ERGAS ↓ | SCC ↑ | Q ↑ | $D_\lambda$ ↓ | $D_S$ ↓ | QNR ↑ |
|---|---|---|---|---|---|---|---|---|---|
| SFIM | 36.9060 | 0.8882 | 0.0318 | 1.7398 | 0.8128 | 0.4349 | 0.0691 | 0.1312 | 0.8109 |
| GS | 37.2260 | 0.9034 | 0.0309 | 1.6736 | 0.7851 | 0.4211 | 0.0397 | 0.1214 | 0.8445 |
| Brovey | 37.7974 | 0.9026 | 0.0218 | 1.3720 | 0.6446 | 0.3857 | 0.0905 | 0.1443 | 0.779 |
| IHS | 38.1754 | 0.9100 | 0.0243 | 1.5336 | 0.6738 | 0.3682 | 0.0418 | 0.1345 | 0.8301 |
| GFPCA | 37.9443 | 0.9204 | 0.0314 | 1.5604 | 0.8032 | 0.3236 | 0.0898 | 0.1815 | 0.7445 |
| PNN | 43.1208 | 0.9704 | 0.0172 | 0.8528 | 0.9400 | 0.739 | 0.0387 | 0.1162 | 0.8494 |
| PANNet | 43.0659 | 0.9685 | 0.0178 | 0.8577 | 0.9402 | 0.7309 | 0.0369 | 0.1219 | 0.8455 |
| MSDCNN | 45.6874 | 0.9827 | 0.0135 | 0.6389 | 0.9526 | 0.7759 | 0.0368 | 0.1112 | 0.8560 |
| SRPPNN | 47.1998 | 0.9877 | 0.0106 | 0.5586 | 0.9564 | 0.7900 | 0.0364 | 0.1087 | 0.8588 |
| GPPNN | 44.2145 | 0.9815 | 0.0137 | 0.7361 | 0.9510 | 0.7721 | 0.0350 | 0.1078 | 0.8612 |
| Ours | **47.2238** | **0.9892** | **0.0102** | **0.5495** | **0.9602** | **0.8026** | **0.0361** | **0.1012** | **0.8665** |

Table 3: Quantitative comparison of nine metrics on the WordView-III dataset. Best results are highlighted by **bold**. ↑ indicates that the larger the value, the better the performance, and ↓ indicates that the smaller the value, the better the performance.

| Methods | PSNR ↑ | SSIM ↑ | SAM ↓ | ERGAS ↓ | SCC ↓ | Q ↑ | $D_\lambda$ ↓ | $D_S$ ↓ | QNR ↑ |
|---|---|---|---|---|---|---|---|---|---|
| SFIM | 21.8212 | 0.5457 | 0.1208 | 8.9730 | 0.6952 | 0.4531 | 0.0448 | 0.1265 | 0.8347 |
| GS | 22.5608 | 0.5470 | 0.1217 | 8.2433 | 0.7131 | 0.4411 | **0.0350** | 0.2011 | 0.7695 |
| Brovey | 22.5060 | 0.5466 | 0.1159 | 8.2331 | 0.7033 | 0.4394 | 0.0481 | 0.2006 | 0.7603 |
| IHS | 22.5579 | 0.5354 | 0.1266 | 8.3616 | 0.6994 | 0.4301 | 0.0356 | 0.2073 | 0.7634 |
| GFPCA | 22.3344 | 0.4826 | 0.1294 | 8.3964 | 0.6987 | 0.3115 | 0.0528 | 0.1214 | 0.8325 |
| PNN | 29.9418 | 0.9121 | 0.0824 | 3.3206 | 0.954 | 0.8679 | 0.046 | 0.0933 | 0.8654 |
| PANNet | 29.6840 | 0.9072 | 0.0851 | 3.4263 | 0.9512 | 0.8631 | 0.0474 | 0.0942 | 0.8634 |
| MSDCNN | 30.3038 | 0.9184 | 0.0782 | 3.1884 | 0.9577 | 0.8763 | 0.0432 | 0.0877 | 0.8732 |
| SRPPNN | 30.4346 | 0.9202 | 0.0770 | 3.1553 | 0.9581 | 0.8776 | 0.0414 | 0.0909 | 0.8719 |
| GPPNN | 30.1785 | 0.9175 | 0.0776 | 3.2593 | 0.9569 | 0.8739 | 0.0438 | 0.0936 | 0.8671 |
| Ours | **30.5425** | **0.9216** | **0.0768** | **3.1049** | **0.9567** | **0.8803** | 0.0412 | **0.0872** | **0.8757** |

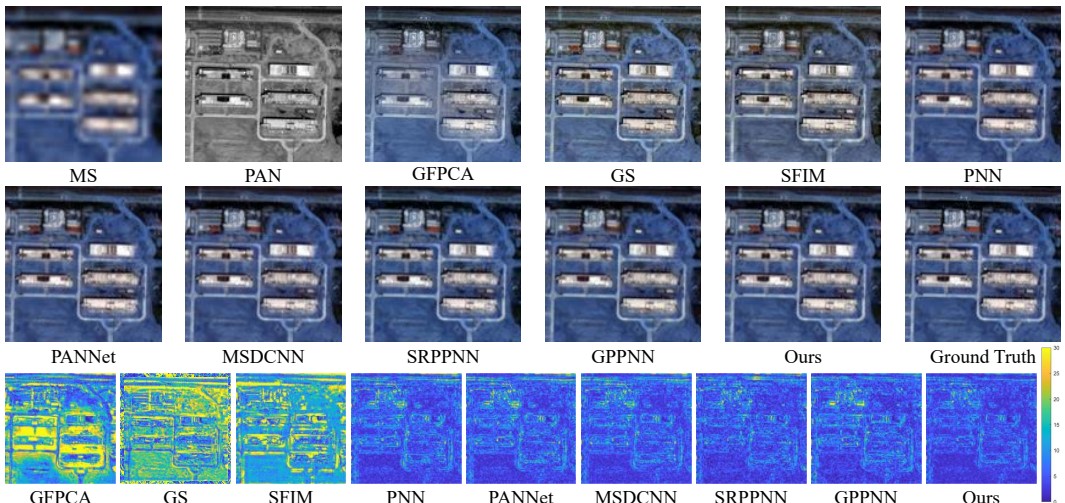

Figure 1: Qualitative visualization comparison of our method with other representative counterparts on a typical satellite image pair from the WordView-II dataset. Images in the last row visualizes the MSE between the fused images and the ground truth.

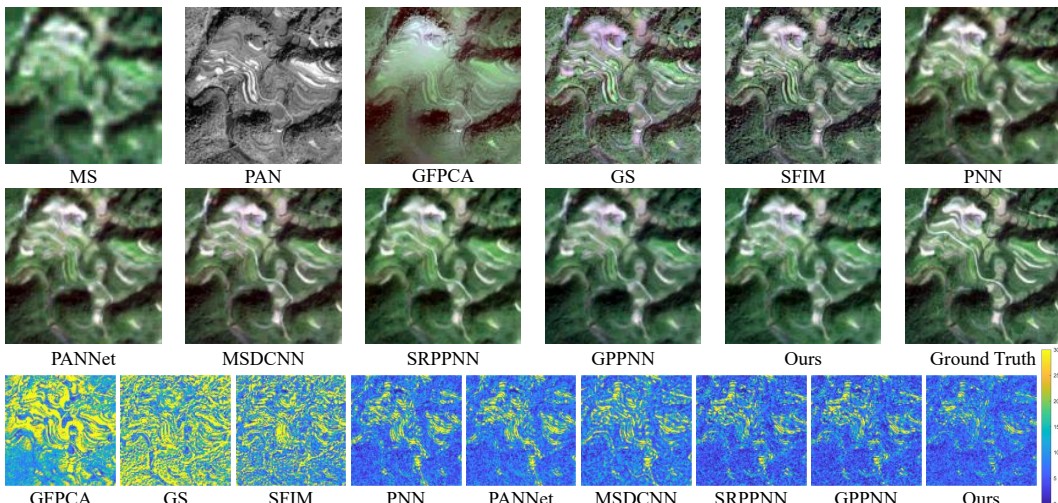

Figure 2: Qualitative visualization comparison of our method with other representative counterparts on a typical satellite image pair from the GaoFen2 dataset. Images in the last row visualizes the MSE between the fused images and the ground truth.

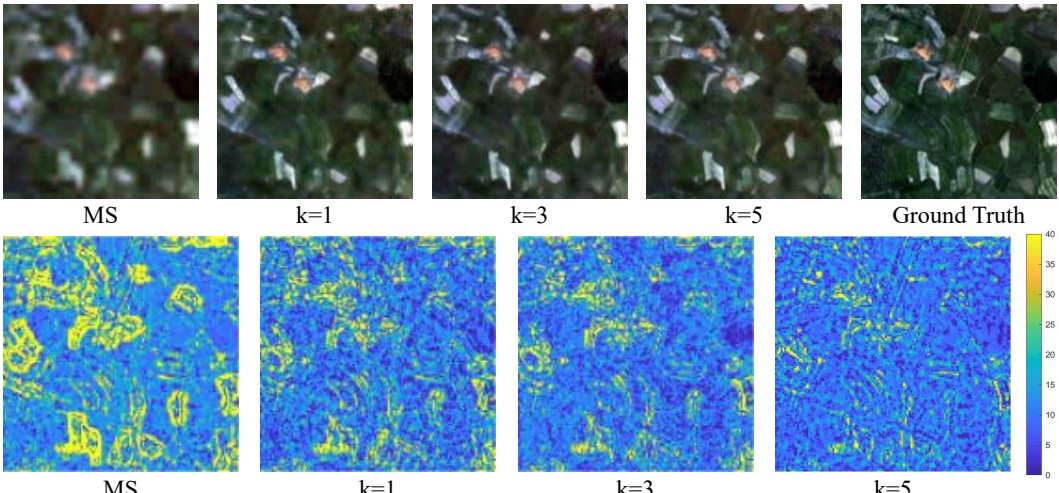

Figure 3: Intermediate visual results of different alternate iterations of ARFNet on WorldView-II. The last row visualizes the MSE residues between the fused results and the ground truth.