# OpenReview forum: "Panchromatic and Multispectral Image Fusion via Alternating Reverse Filtering Network"
_NeurIPS.cc/2022/Conference — NeurIPS 2022 Accept_

### Official Review · Reviewer_T9TL · 2022-07-11

**Rating:** 5
**Confidence:** 4
**Soundness:** 3 good
**Presentation:** 2 fair
**Contribution:** 2 fair

**Summary:**

The manuscript first formulate the pan-sharpening problem as a reverse filtering process and propose an alternating reverse filtering network (ARFNet) to solve it. Besides, the authors develop multi-scale Gaussian convolution modules to obtain better filtering effect. Experimental results on both simulated and real-world scenes show that the proposed ARFNet outperforms other methods.

**Questions:**

1.	As stated in [1], the optimal iteration number is different for variant filters and needs to be set empirically. Is the iteration number set equally in the ablation study of different initialization methods?

2.	The learned kernels are close to the initialized Gaussian kernels as stated in Line 241, can the learned kernels gain large performance improvement compared with the initialized ones?

3.	The authors cited the recent method[2], but not compared the experimental results with it.

4.	How about the computation cost and efficiency compared with other methods?

5.	It’s better to include related source citation for the definition, the theorem and reverse filtering formulation.

6.	Please check the subtraction and addition symbols in Fig. 1, they are a bit confusing.

[1] Xin Tao, Chao Zhou, Xiaoyong Shen, Jue Wang, and Jiaya Jia. Zero-order reverse filtering. In 2017 IEEE International Conference on Computer Vision (ICCV), pages 222–230, 2017.
[2] Man Zhou, Xueyang Fu, Jie Huang, Feng Zhao, Aiping Liu, and Rujing Wang. Effective pan-sharpening with transformer and invertible neural network. IEEE Transactions on Geoscience and Remote Sensing, 60:1–15, 2022.


**Limitations:**

There is no potential negative societal impact mentioned. Considering the paired dataset is expensive to obtain, the authors could also explore methods in unsupervised manner.

**Strengths And Weaknesses:**

Strengths:
1.	This manuscript combines the deep unrolling model and the reverse filter process, and proposes a novel alternating update deep convolution network for pan-sharpening task.
2.	Experimental results show that this manuscript set up some new state-of-the-art results in both simulated and real-scene datasets.
3.	The writing is clear and easy to understand.
Weakness:
1.	The reverse filter method has been already applied in super-resolution, deconvolution in [1], the Gaussian filter and its convergence condition has also been analyzed. Overall, the novelty is limited.

---

> ### Author Response · Authors · 2022-08-02
> **Reponse to Reviewer T9TL**
>
> We appreciate the positive and constructive comments, including novel alternating update deep convolution network, the state-of-the-art results of our method, and clear writing and easy to follow. The raised concerns are addressed as follows.
>
> **About the the iteration number set in the ablation study.**
>
> In the ablation study of different initialization methods, we set the iteration number to 5 for a fair comparison. For the baseline models, we change only one parameter or component, and keep the other parameters and components the same.
>
> **About the performance improvement from the learned kernels.**
>
> Compared with the model that is initialized by fixed Gaussian kernels (III), our model with the learned kernels (ll) can gain large performance improvement. The comparison results are listed as follows:
>
> | Methods  | PSNR↑ | SSIM↑ | SAM↓ | ERGAS↓ | SCC↑  |  Q↑ | Dλ↓| Ds↓| QNR↑|
> | :-----: | :----: | :----: | :----: | :----: | :----: | :----: | :----: | :----: | :----: |
> | (ll) | 41.7587 | 0.9691 | 0.0229 | 0.9540 | 0.9749 | 0.7731 | 0.0631 | 0.1184 | 0.8285 |
> | (lll) | 40.4161 | 0.9612 | 0.0261 | 1.0668 | 0.9667 | 0.7316 | 0.0684 | 0.1275 | 0.8123 |
>
> Although the learned kernels are close to the initialized Gaussian kernels, a large number of multi-scale Gaussian kernels can still bring performance improvements. We will add these experimental results in the revised version.
>
> **About the recent method.**
>
> Because the related work mentioned by reviewer have not released the code when we submitted the paper, there is no comparison with this method in our paper.
>
> **About the computation cost and efficiency.**
>
> The floating-point operations (FLOPs) and the number of parameters of the compared models are provided in the following table:
>
> | Methods  | PNN | PANNet | MSDCNN | SRPPNN | GPPNN | Ours |
> | :-----: | :----: | :----: | :----: | :----: | :----: | :----: |
> | Flops | 1.13 | 1.12 | 3.92 | 21.11 | 1.40 | 2.58 |
> | #Params | 0.69 | 0.69 | 2.39 | 17.11 | 1.12 | 1.66 |
>
> As can be seen, PNN and PANNet have the fewest FLOPs and parameters, but the performance is the worst. SRPPNN exhibits large increases in the number of parameters and FLOPs as the number of convolutional layers and the complexity of the network design rise. Notably, it achieves the second highest performance at the expense of massive model computation and storage. Additionally, the most comparable solution to ours, GPPNN, is organized around the model-based unrolling principle and has comparable model parameters and flops reductions but inferior performance. In summary, our network achieves a favorable trade-off between calculation, storage, and model performance compared with other methods.
>
> **About the related citation for definition and theorem.**
>
> Take the reviewer's suggestions into account, we will add the related source citation for the definition, the theorem and reverse filtering formulation. Specifically, Definition 3.2 and Theorem 3.2 can be found in reference [*1].
>
> [*1] Banach, S, “Surles operations dans ensembles abstraits etleur application aux equations integrales,” Fundamenta Mathematicae, 3. 51-57. 1922.
>
> **About the subtraction and addition symbols.**
>
> Thanks for your careful reading and suggestions.The position of the legend about subtraction and addition is reversed. We will revise it in a revision.
>
> **About the novelty.**
>
> Reverse filtering is a classical algorithm, which inspired a lot of works. However, our approach is significantly different from classical reverse filtering. To be specific, (1) the classical reverse filtering can only solve the inverse problems of single image such as denoising, deblurring and super-resolution. Our approach tailors the classical reverse filtering in an alternating iteration manner based on the theortically feasible fixed point equation system for multispectral image fusion. As mentioned in our paper, with such theoretical support, the proposed algorithm can also be extended to other image fusion tasks that involve multiple images. (2), the classical reverse filtering needs to select a suitable and fixed filter function, such as adaptive manifold filter, rolling guidance filter, wlsFilter and so on. For this problem, we introduced the multi-scale Gaussian kernels to approximate arbitrary functions and combine them into deep networks. By doing so, the data-driven deep neural networks can help our model to obtain an appropriate function rather than a fixed and manually defined one, improving the model’s generalization and applicability.

---

> > ### Comment · Reviewer_T9TL · 2022-08-08
> > **Thank you for your response**
> >
> > I would like to thank the authors for their response. Some questions are addressed properly, including the efficiency evaluation and the comparison between initialized kernels and trained kernels, which improve the completeness.
> >
> > However, the claimed main contribution that solving the multiple image fusion problem as a reverse filtering process is limited. Although the authors argued that they tailor the classical reverse filtering in an alternating iteration manner, it’s much closer to a reasonable implementation rather than an inspiring innovation. Besides, the stability and convergence of the iterative process totally benefit from the classical reverse filtering. Considering all the related demonstration have been given thoroughly in “Zero-order Reverse Filtering”, the authors oversell the theoretical contribution.
> >
> > Overall, I feel the work is a good engineering approach, but the novelty is a bit limited.
> >
> > Therefore I will keep my score unchanged.

---

### Official Review · Reviewer_X4rs · 2022-07-11

**Rating:** 4
**Confidence:** 5
**Soundness:** 3 good
**Presentation:** 3 good
**Contribution:** 2 fair

**Summary:**

The authors propose an alternating reverse filtering network for pan-sharpening. Specifically, a learnable multi-scale Gaussian kernel module is designed to ensure the convergence of the iterative process. The experiments on diverse scenes show that the proposed network achieves better results compared to other pan-sharpening methods.

**Questions:**

How about the parameter numbers and inference time for the proposed network and competing networks?

**Limitations:**

Yes

**Strengths And Weaknesses:**

Strength:
The paper is well-written. The experiments show that the method can perform well in terms of quantitative analysis and qualitative comparison.

Weakness:
1. Originality
The originality of this work is limited. The reverse filtering is derived from [23] and the fusion network is derived from [63]. The description in 3.2, 3.3, and 3.4 sections looks quite similar to [23].
2. Clarity
-The multi-scale Gaussian convolution is a group of convolutional layers initialized by 2D Gaussian kernels, but the detailed architecture of these layers is missing. What are the exact architectures of g and f?
-The contribution states that ‘In contrast to existing model-driven methods, our iterative network can obtain HR MS without the need for pre-defined exact priors or assumptions’. It is confusing and more explanations are needed. What does the pre-defined exact prior mean in existing model-driven methods? How does the proposed method obtain HR MS without the need for pre-defined exact prior, compared to prior methods?
3. Quality
-In Table2, the MSDCNN achieves 0.8251 in QNR and the result of proposed method is 0.8236. However, the result 0.8236 is in bold instead of 0.8251. The proposed method only achieves the best result in one metric.
-Observed from Figure 3, the Brovey method achieves better fusion results in spatial texture compared to proposed method.

---

> ### Author Response · Authors · 2022-08-02
> **Reponse to Reviewer X4rs (part 2/2)**
>
> **About the Quality.**
>
> 1)Sorry for our carelessness. When transcribing the data to Table 2, I inverted the last two digits of the data (The 0.8251 should have been 0.8215.) . We will correct this in a revised version. The proposed method still achieves the best result in two metrics in bold. In addition, according to the calculation formula of QNR and the relationship among QNR, Ds and D_lamda [*1], we can tell that the QNR of MSDCNN should not have such a high value (0.8251).
>
> 2)In the visualization results of the Brovey method in Figure 3, there is obvious spectral distortion, which is because the CS-based methods replace the separated spatial component with the PAN image. Therefore, the results of Brovey look more like the PAN image in spatial texture. However, such spectral distortion is unacceptable in this task. In general, our method achieves better performance than the Brovey methods.
>
> [*1] Multispectral and panchromatic data fusion assessment without reference. Photogrammetric Engineering & Remote Sensing, 74(2):193–200, 2008.
>
> **About the parameter numbers and inference time.**
>
> The floating-point operations (FLOPs) and the number of parameters of the compared models are provided in the following table:
>
> | Methods  | PNN | PANNet | MSDCNN | SRPPNN | GPPNN | Ours |
> | :-----: | :----: | :----: | :----: | :----: | :----: | :----: |
> | Flops | 1.13 | 1.12 | 3.92 | 21.11 | 1.40 | 2.58 |
> | #Params | 0.69 | 0.69 | 2.39 | 17.11 | 1.12 | 1.66 |
>
> As can be seen, PNN and PANNet have the fewest FLOPs and parameters, but the performance is the worst. SRPPNN exhibits large increases in the number of parameters and FLOPs as the number of convolutional layers and the complexity of the network design rise. Notably, it achieves the second highest performance at the expense of massive model computation and storage. Additionally, the most comparable solution to ours, GPPNN, is organized around the model-based unrolling principle and has comparable model parameters and flops reductions but inferior performance. In summary, our network achieves a favorable trade-off between calculation, storage, and model performance compared with other methods.

---

> ### Author Response · Authors · 2022-08-02
> **Reponse to Reviewer X4rs (part 1/2)**
>
> We thank reviewer for the feedback. We are encouraged that the reviewer confirms the good writing of our paper and convincing and sufficient experiments. The raised concerns are addressed as follows.
>
> **About originality of this work is limited.**
>
> 1)Reverse filtering is a classical algorithm, which inspired a lot of works. As the research background, the introduction of reverse filtering is provided, to make readers better understand the background of reverse filtering. However, our approach is significantly different from classical reverse filtering. To be specific, (1) the classical reverse filtering can only solve the inverse problems of single image such as denoising, deblurring and super-resolution. Our approach tailors the classical reverse filtering in an alternating iteration manner based on the theortically feasible fixed point equation system for multispectral image fusion. As mentioned in our paper, with such theoretical support, the proposed algorithm can also be extended to other image fusion tasks that involve multiple images. (2), the classical reverse filtering needs to select a suitable and fixed filter function, such as adaptive manifold filter, rolling guidance filter, wlsFilter and so on. For this problem, we introduced the multi-scale Gaussian kernels to approximate arbitrary functions and combine them into deep networks. By doing so, the data-driven deep neural networks can help our model to obtain an appropriate function rather than a fixed and manually defined one, improving the model’s generalization and applicability.
>
> 2)In addition, our fusion network is quite different from [63]. (1), the model solving processes between our network and [63] are quite different. [63] employs the gradient projection method to solve their assumptions with penalized optimization. The penalty term is approximated by a proximal operator which is set empirically without constraints. The whole process is constrained in the following steps. At each iteration, it is necessary to perform down-sampling and blurring to obtain the residual map. And then the residual map is added back to the H by inverse down-sampling approximate operation. In comparison, in our approach, there are no penalized priors in the solution process of the fixed-point equation system, and we do not need to design various approximation operators. (2), as both networks are interpretable, we can see that the interpretation represented by the various parts of the two networks is also different. For example, the convolutional operators in the MS Block [63] represent the down-sampling matrix, circular convolution matrix, transpose matrix approximation operator and proximal operator corresponding to penalty. However, there is only a multi-scale Gaussian convolution module in our model that represents the low-passing filter.
>
> **About Clarity.**
>
> 1)As shown in Figure 1, network g(•) and network f(•) have the same structure that is based on the multi-scale Gaussian convolution module. The multi-scale Gaussian convolution module consist of a series of Group Convolutions which are already widely used in previous works [*1, *2]. We use Gaussian kernels to initialize the kernels in Group Convolutions. The code in PyTorch is implemented as ```F.conv2d(x, Gaussian_kernels, padding, groups=input_channels)```. The sizes of Gaussian kernels are different, so there are different Group Convolutions. The outputs of different Group Convolutions are added, and the combined coefficients are learnable.
>
> 2)Take the work we mentioned in the related work as an example, the CSC model [*3] is based on a strong assumption that there are common and private features between PAN and MS that can be represented by the dictionary. The two types of features can be expressed independently is the pre-defined exact prior in the CSC model. In fact, the two types of features cannot be completely separated. Solving such ideal assumptions may cause system model error. Similarly, the reviewer mentions [63] establishes a pre-defined image generation process as priors: P=SH，L=DKH. Based on these priors, the model needs to be obtained transpose DK during the solution process. But the DK and transpose DK cannot get a strict correspondence in the process of optimization. In contrast, our approach does not explicitly establish these priors that avoids the dependency on pre-defined exact priors. Additionally, our approach does not need to involve the calculation of transpose DK or the pseudo inverse DK in the process of solving the fixed-point equation system to obtain HRMS.
>
> [*1] ImageNet classification with deep convolutional neural networks, Communications of the ACM, pp. 84-90, 2017.
>
> [*2] "ShuffleNet: An Extremely Efficient Convolutional Neural Network for Mobile Devices," IEEE/CVF Conference on Computer Vision and Pattern Recognition, 2018.
>
> [*3] "PanCSC-Net: A Model-Driven Deep Unfolding Method for Pansharpening," in IEEE Transactions on Geoscience and Remote Sensing, 2022.

---

### Official Review · Reviewer_vNgg · 2022-07-11

**Rating:** 7
**Confidence:** 4
**Soundness:** 4 excellent
**Presentation:** 3 good
**Contribution:** 3 good

**Summary:**

This paper proposes an alternating reverse filtering network for multi-spectral image fusion. Motivated by classical reverse filtering, the authors develop an unrolling network based on alternating fixed-point iterations and the learnable multi-scale Gaussian kernel module, which is interpretable. Unlike existing model-driven methods, the proposed iterative network avoids the dependency on pre-defined exact priors and its convergence can be guaranteed theoretically. In addition, this paper introduces a new perspective to solving the image fusion problems. Extensive experiments on diverse scenes to thoroughly verify the performance of the proposed method.

**Questions:**

1. The structure loss $L_s$ is based on structural similarity, what will happen if loss function $L_s$ uses the L2 loss function?
2. If each alternate iteration output is supervised with losses, would it be better?
3. In the ablation study about the number of iterations K, why does the performance seem to decrease slightly when K reaches 7 compared with K=5? What happens to the model's performance in the case of the larger K?
4. As far as I know, the GPPNN network compared in the manuscript is also based on an unrolling method. What is the K used in GPPNN here? If the settings of K are different, how to ensure fair comparisons of the two unrolling methods?
5. As mentioned in the Weaknesses, the authors are suggested to provide specific parameters for initializing the Gaussian kernel.

**Limitations:**

The limitations and potential negative social impact of their work have been addressed.

**Strengths And Weaknesses:**

Strengths:
1. The authors innovatively propose a new multi-spectral image fusion method without requiring well-designed priors. The exploration of tailoring classic methods for multi-spectral image fusion and combining them with deep networks is worth trying and encouraging.
2. In the image fusion community, the idea that formulating pan-sharpening as an alternately iterative reverse filtering process is interesting. Other researchers could utilize it for further exploration in the field of image fusion.
3. The learnable multi-scale Gaussian kernel module is well designed and the convergence of the iterative process is sufficiently supported by theoretical analysis.
4. Quantitative and qualitative results clearly verify the effectiveness and scalability of the proposed method.

Weaknesses:
1. More ablation results about the loss functions and iterations should be provided. Please refer to the detailed Questions part.
2. The parameters such as sigma for initializing the Gaussian kernel should be given in line 239.

---

> ### Author Response · Authors · 2022-08-02
> **Reponse to Reviewer vNgg**
>
> We appreciate the positive and constructive comments, including interesting ideas, a new method, a well-designed module with sufficient theoretical analysis, and convincing quantitative and qualitative experiments. The raised concerns are addressed as follows.
>
> ** About the structure loss function.**
>
> Thank you for the insightful question. We have already conducted some experiments to explore how to design the loss function for the multispectral image fusion problem at the very beginning of the research. The comparison experiment mentioned by the reviewer also has been done before. For the story flow of the paper, we did not provide these experiments in the paper. As suggested, we provide the experimental results in the table below, where the configurations (I) and (II) are the loss Ls using structural similarity loss and L2 loss, respectively.
>
> | Config  | PSNR↑ | SSIM↑ | SAM↓ | ERGAS↓ |
> | :-----: | :----: | :----: | :----: | :----: |
> | (l) | 41.7587 | 0.9691 | 0.0229 | 0.9540 |
> | (ll) | 41.7913  | 0.9677 | 0.0234 | 0.9569 |
>
> As presented in this table, if the loss function Ls uses the same L2 loss function as the reconstruction loss, the performance in terms of the PSNR metric will be improved, but the other metrics (such as SSIM) will be decreased slightly. Based on the comprehensive consideration, we choose to use the structure loss Ls based on the structural similarity. In addition, we found that the best results could be obtained when both structure loss and L2 loss are used. However, in this case, the hyperparameter needs to be carefully adjusted. Hence, we did not adopt such a combination strategy.
>
> ** About supervision at each alternate iteration output.**
>
> In the previous works[*1-*3], the loss function of the training process is usually supervised at the output of an unrolling-based deep network. To make a fair comparison, we did not add the supervision at each alternate iteration output. To answer the reviewer’s question, we carried out the experiments following the same strategy of the unrolling-based work[*4], in which each alternate iteration output is supervised with losses. The experimental results on the WordView-II dataset are presented in the following table.
>
> | Config  | PSNR↑ | SSIM↑ | SAM↓ | ERGAS↓ |
> | :-----: | :----: | :----: | :----: | :----: |
> | (l) | 41.7587 | 0.9691 | 0.0229 | 0.9540 |
> | (ll) | 41.7843  | 0.9698 | 0.0228 | 0.9523 |
>
> Configuration (I) is the original model configuration while Configuration (II) represents that the loss is supervised at the output of each alternate Iteration. As the table shows, the losses are supervised at the output of each alternate iteration, giving the model a performance boost.
>
> [*1] Denoising prior driven deep neural network for image restoration. IEEE transactions on pattern analysis and machine intelligence, 41(10):2305–2318, 2018.
>
> [*2] Pancsc-net: A model-driven deep unfolding method for pansharpening. IEEE Transactions on Geoscience and Remote Sensing, pages 1–13, 2021.
>
> [*3] Deep gradient projection networks for pan-sharpening. In IEEE Conference on Computer Vision and Pattern Recognition, pages 1366–1375, June 2021.
>
> [*4] Memory-augmented deep unfolding network for com409 pressive sensing. In ACM International Conference on Multimedia (ACM MM), 2021.
>
> ** About the number of iterations K.**
>
> Observing the results from Table 6 of the main paper, the model’s performance has improved considerably as the number of stages increases until reaching 5. When increasing the number K, there is still a slight increase in the performance in terms of some metrics (such as PSNR). Due to limited time, we only added one experiment (that is, K=8), and present the experimental results in the table below:
>
> | K | 5 | 6 | 7 | 8 |
> | :-----: | :----: | :----: | :----: | :----: |
> | PSNR↑ | 41.7587 | 41.7614 | 41.7603 | 41.6671 |
> | SSIM↑ | 0.9691  | 0.9690 | 0.9688 | 0.9680 |
>
> As shown in this table, the results show that a descending trend, which may be caused by the difficulty of gradient propagation. Therefore, we set K to 5 as default to balance the performance and computational complexity.
>
> ** About the K used in GPPNN.**
>
> In the work [*3], the authors demonstrate that the GPPNN model can achieve the best performance when the value of K is set to 8. Therefore, in our comparison experiments, the GPPNN model is also set with K=8. To ensure a fair comparison of the two unrolling models, we use the best parameter settings of each model separately, instead of using the same value of K. Specifically, for the GPPNN model, the K is set to 8, and our model adopts K=5.
>
> ** About the parameters for initializing the Gaussian kernel.**
>
> In our implementation, the parameters of the initialized Gaussian kernel vary with the sizes of the kernel. The sigma of the Gaussian function is set to ks/4, where ks denotes the kernel size. We will add the details of the parameters for initializing the Gaussian kernel in the revised version.

---

### Meta-Review · Area_Chair_SBiQ · 2022-08-30

**Recommendation:** Accept
**Confidence:** Less certain

**Metareview:**

The paper presents a pan sharpening image fusion approach using deep learning. The overall review sentiment leaned towards accepting the paper. The reviewers appreciated the reformulation of the problem as an iterative reverse filtering process and thought the technique was generalizable, broadening its potential impact. Some concern was mentioned that the paper directly adapts existing techniques, and the theory is directly pulled from those papers. In that sense the paper is applied. The experimental results were convincing to the reviewers in general, which helps justify the mostly applied nature of the paper.

**Award:**

No

---

### Decision · Program_Chairs · 2022-09-14

Accept